# Unveiling the Lewis Acid Catalyzed Diels–Alder Reactions Through the Molecular Electron Density Theory

**DOI:** 10.3390/molecules25112535

**Published:** 2020-05-29

**Authors:** Luis R. Domingo, Mar Ríos-Gutiérrez, Patricia Pérez

**Affiliations:** 1Department of Organic Chemistry, University of Valencia, 46100 Burjassot, Valencia, Spain; rios@utopia.uv.es; 2Department of Chemistry and Chemical Biology, McMaster University Hamilton, Hamilton, ON L8S 4L8, Canada; 3Department of Chemistry, Computational and Theoretical Chemistry Group, Faculty of Sciences, University Andres Bello, 8370146 Santiago, Chile; p.perez@unab.cl

**Keywords:** Diels–Alder, Lewis acid, catalysis, molecular electron density theory, relation mechanism

## Abstract

The effects of metal-based Lewis acid (LA) catalysts on the reaction rate and regioselectivity in polar Diels–Alder (P-DA) reactions has been analyzed within the molecular electron density theory (MEDT). A clear linear correlation between the reduction of the activation energies and the increase of the polar character of the reactions measured by analysis of the global electron density transfer at the corresponding transition state structures (TS) is found, a behavior easily predictable by analysis of the electrophilicity ω and nucleophilicity *N* indices of the reagents. The presence of a strong electron-releasing group in the diene changes the mechanism of these P-DA reactions from a *two-stage one-step* to a two-step one via formation of a zwitterionic intermediate. However, this change in the reaction mechanism does not have any chemical relevance. This MEDT study makes it possible to establish that the more favorable nucleophilic/electrophilic interactions taking place at the TSs of LA catalyzed P-DA reactions are responsible for the high acceleration and complete regioselectivity experimentally observed.

## 1. Introduction

The Diels–Alder (DA) reaction between a conjugated diene and an ethylene derivative to yield a cyclohexene, reported for the first time by Diels and Alder in 1928 [1], is one of the most studied organic reactions from a synthetic as well as a theoretical viewpoint (see Scheme 1) [2,3]. Its usefulness arises from its versatility and its remarkable selectivity, many synthetic routes to cyclic compounds are made possible through DA reactions, which can involve a large variety of dienes and ethylene derivatives.

The DA reaction between butadiene **4** and ethylene **5**, classified by Woodward and Hoffmann as a “pericyclic” reaction [4] was chosen as the paradigm of DA reactions (see Scheme 2). However, while the DA reaction between cyclopentadiene (Cp, **1**) and maleic anhydride **2** takes place easily a room temperature, the DA reaction between butadiene **4** and ethylene **5** does not take place in the laboratory. It must be forced to take place: after 17 h at 165 °C and 900 atmospheres, it gives a 78% yield [5]. The Frontier Molecular Orbital [6] (FMO) theory was used to study the reactivity of DA reactions [7,8]. Concepts such as highest occupied molecular orbital (HOMO)–lowest unoccupied molecular orbital (LUMO) energy gaps and molecular orbital (MO) interactions between the interacting molecules were used within the FMO to explain the reactivity of DA reactions [9].

After an exhaustive study of organic reactions carried out over 20 years in 2016, Domingo proposed the Molecular Electron Density Theory [10] (MEDT) for the study of the organic reactivity. This theory proposes that the changes of electron density along an organic reaction, and no MO interactions such as the FMO theory proposed [6], are responsible for the chemical reactivity of organic molecules. A recent MEDT study of the DA reaction between butadiene **4** and ethylene **5** showed that the bonding changes along the reaction path take place sequential and symmetrically, and in a non-concerted (simultaneous) cyclic arrangement, thus ruling out the “pericyclic mechanism” [11].

A DFT study of the DA reactions of Cp **1** and the series of cyanoethylenes, experimentally reported by Sauer in 1964 [12], allowed establishing a linear relationship between the logarithm of the experimental rate constants and the polar character of DA reactions, measured through the global electron density transfer [13] (GEDT) at the transition state structures (TSs), R^2^ = 0.99, making it possible to establish the polar Diels–Alder (P-DA) mechanism in 2009 [14]. A very good correlation between the activation barriers and the GEDT for the DA reactions of Cp **1** with a series of substituted ethylenes of increased electrophilic character was established, R^2^ = 0.89 (see Figure 1) [14]. The analysis of the electrophilic [15] ω and nucleophilicity [16] *N* indices defined within the conceptual DFT [17,18] (CDFT) allow stablishing the feasibility of P-DA reactions.

Thus, while the DA reaction between Cp **1** and maleic anhydride **2** reported by Diels and Alder takes place easily a room temperature [1], the reaction between Cp **1** and acrolein **7** only proceeds at higher temperatures. As show Figure 1, the reaction with acrolein **7** was the poorest activated P-DA reaction of this series. However, coordination of the carbonyl oxygen of acrolein **7** to the B or Al centers of the BH**_3_** and AlCl**_3_** Lewis acids (LAs) decreased considerably the activation barriers through more polar processes. This finding was in complete agreement with the increase of the calculated GEDT at the corresponding TSs: 0.14 e (acrolein **7**), 0.27 e (BH**_3_** LA complex **7-BH_3_**), and 0.37 e (AlCl**_3_** LA complex **7-AlCl_3_**), a behavior that correlated very well with the increase of the electrophilicity ω index of these ethylene derivative; 1.84 eV (acrolein **7**), 3.24 eV (BH**_3_** LA complex **7-BH_3_**) and 4.62 eV (AlCl**_3_** LA complex **7-AlCl_3_**), R^2^ = 1.00 (see Figure 2). It was proposed that “our reactivity model accurately accounts for the effects of the LA on the P-DA reactions. LA-catalyzed DA reactions experience a high acceleration as a consequence of the increase in the electrophilicity of the carbonyl compound, which favors the DA reaction through a more polar process” [14,19].

Recently, the P-DA reactions of nucleophilic dienes with electrophilically activated ethylenes, such as the aforementioned series of P-DA reactions, have been classified as reactions of forward electron density flux (FEDF) [20]. However, there are heterodienes such as tetrazine **8** (see [Fig molecules-25-02535-ch001]) and nitroethylene **9** that participate as strong electrophiles in P-DA reactions. These type of DA reactions, which demands the nucleophilic activation of the ethylene to the reaction takes place, have been classified as the reverse electron density flux (REDF) [20].

LA catalysts considerably extend the useful scope of the DA reaction, enhancing the reaction rate and leading to significant changes in *endo/exo* stereo- and regioselectivities in comparison with the unanalyzed process [21,22]. In addition, the use of chiral ligands coordinated to the metal of LAs has acquired a great significance in the asymmetric synthesis of pertinent natural and pharmaceutical compounds.

The LA catalyzed DA reactions between Cp **1** and acrolein **7** present in Figure 1 are P-DA reactions of the FEDF, but also there are LA catalyzed DA reactions of REDF. In 1999, the LA catalyzed P-DA reactions of nitroethylene **9** with three ethylenes **10** of increased nucleophilic character was studied (see Scheme 3) [19]. Coordination of the BH**_3_** LA to a one oxygen of nitroethylene **9** considerable reduced the activation energies of the non-catalyzed P-DA reactions to notably increase the GEDT calculated at the corresponding TSs. The presence of the LA catalyst considerably increased the *endo* stereoselectivity and the regioselectivity of these P-DA reactions. Inclusion of solvent effects changed the mechanism of the LA catalyzed DA reaction with the strongly nucleophilic dimethylvinylamine **10** (R = NMe**_2_**) from one-step to a two-step mechanism, as a consequence of the stabilization of the corresponding zwitterionic intermediate. In this earlier study, Domingo proposed “The role of the Lewis acid catalyst can be understood as an increase of the electrophilic character of the nitroethene due to a stabilization of the corresponding TS through a delocalization of the negative charge that is being transferred along the nucleophilic attack of the substituted ethylene” [19].

In an Angewandte Chemie communication, in 2020, entitled “How Lewis Acids Catalyze Diels–Alder Reactions”, Bickelhaupt proposed a “novel physical mechanism” based on his Activation Strain Model [23] (ASM) [24]. He proposed that LA catalysts accelerate DA reactions by diminishing the Pauli repulsions rather that the proposed “enhanced the donor–acceptor [HOMO_diene_–LUMO_dienophile_] interactions” within the FMO theory. However, ASM has significant chemical incoherencies. ASM is based on the MO energy decomposition analysis scheme proposed by Morokuma in 1981 [25], and consequently it has no significance within the Hohenberg and Kohn DFT framework [26], as Kohn–Sham orbitals do not define any wave function such as the Hartree–Fock formalism, although they are used to calculate the one-electron density distribution function [27]. Thus, the energy decomposition analysis of the interaction energies into electrostatic interactions, orbital interactions and Pauli repulsions have any chemical meaning within the DFT as this analysis has a MO contribution.

Chemical concepts as nucleophile and electrophile, established in the first quarter of the last century, are commonly used in Organic Chemistry to describe the chemical organic reactivity. After the establishment in 2009 of the mechanism in P-DA reactions, these concepts are widely used to describe the chemical reactivity in DA reactions [18]. Herein, a comprehensive MEDT study about the role of the metal-based LA catalysts in P-DA reactions of FEDF is reported. To this end, the P-DA reactions of Cp **1** with five LA:acrolein complexes of increased electrophilic character, and the LA catalyzed DA reactions of three cyclopentadiene derivatives **12**–**14** of increased nucleophilic character with the complex **7-BF_3_** are studied (see Scheme 4). The main propose of this MEDT study is to shed light on all mechanistic details of these pertinent organic reactions which have acquired a great significance in the asymmetric synthesis with the use of chiral LA catalyst.

## 2. Results and Discussion

The present MEDT study has been divided in eight sections: i) in Section 2.1, an analysis of the global and local CDFT reactivity indices at the ground state (GS) of the reagents is performed; ii) in Section 2.2, an analysis of the effects of the LAs and the substitution on Cp **1** in the electronic structure of the reagents is carried out; iii) in Section 2.3, the P-DA reactions between Cp **1** and the series of LA complexes are studied; iv) in Section 2.4, P-DA reactions between substituted Cps **12** - **14** and complex **7-BF_3_** are studied; v) in Section 2.5, the solvent effects in P-DA reactions between Cp **13** and complex **7-BF_3_** are evaluated; vi) in Section 2.6, the thermodynamic parameters of the P-DA reactions between Cp **13** and complex **7-BF_3_** are discussed; vii) in Section 2.7, the bonding changes along the P-DA reaction between Cp **1** and complex **7-BF_3_** are analyzed; and finally, and viii) in Section 2.8, a comparative electron localization function (ELF) topological analysis between *two-stage one-step* and two-step mechanisms in LA catalyzed DA reactions is carried out.

### 2.1. Analysis of the Global and Local CDFT Reactivity Indices at the GS of the Reagents

First, in order to understand the participation of the dienes and the LA complexes in these P-DA reactions, an analysis of the CDFT indices at the GS of the reagents, computed at the B3LYP/6-31G(d) level, was performed [18]. The global CDFT indices, namely, electronic chemical potential μ, chemical hardness η, global electrophilicity ω, and global nucleophilicity *N*, of the reagents are gathered in Table 1.

The electronic chemical potential [28,29] μ of Cp **1** and the Cp derivatives **12**–**14**, between −3.01 (**1**) and −2.09 (**14**) eV, are higher than those of acrolein **7**, μ = −4.38 eV, and the LA complexes, −4.61 (**7-AlMe_3_**) and −6.20 (**7-BF_3_**) eV, indicating that the GEDT in these P-DA reactions will take place from these Cp derivatives towards acrolein **7** and these LA complexes. Thus, these reactions are classified as the FEDF.

The electrophilicity [15] ω and nucleophilicity [16] *N* indices of ethylene **5** are 0.73 and 1.87 eV, respectively, being classified as marginal electrophile and marginal nucleophile within the electrophilicity [30] and nucleophilicity [31] scales. These behaviours make that ethylene **5** does not participate in P-DA reaction.

The electrophilicity ω and nucleophilicity *N* indices of acrolein **7** are 1.84 and 2.13 eV, being classified as strong electrophile and moderate nucleophile. Inclusion of the electron-withdrawing (EW) carbonyl –CHO group in ethylene **5** markedly increases the electrophilicity of acrolein **7**, allowing its participation in P-DA reactions as an electrophilic ethylene. A more remarkable effect is observed with the coordination of a LA to the oxygen atom of acrolein **7** and the electrophilicity ω index of these LA complexes rages from 3.20 (**7-BH_3_**) to 4.61 (**7-AlCl_3_**) eV. A clear correlation between the LA character of the LA salts and the increase of the electrophilic character of the corresponding LA complexes can be established. All these complexes have electrophilicity ω indices higher than 3.0 eV, accounting for the strong activation of acrolein **7**. In general, the electrophilicity ω index of these LA complexes depends on the metal, increasing in the order B < Ti < Al. On the other hand, the counter ion of the LA salt has also a relevant role; thus, the electrophilicity ω index of **7-BF_3_**, 3.29 eV, and **7-AlCl_3_**, 4.61 eV, are higher than that of **7-BH_3_**, 3.20 eV, and **7-AlMe_3_**, 3.60 eV, respectively.

In general, a decrease of the nucleophilicity *N* indices of these LA complexes with the increase of its electrophilic character is observed. They have nucleophilicity *N* indices lower than 2.0, being classified as marginal nucleophiles. An unexpected value is found at of **7-AlMe_3_**, *N* = 3.05 eV, which is classified as a strong nucleophile.

The electrophilicity ω and nucleophilicity *N* indices of Cp **1** are 0.83 and 3.37 eV, respectively, being classified in the borderline of moderate electrophiles and as strong nucleophile. Consequently, Cp **1** will participate in P-DA reactions as a strong nucleophile without any nucleophilic activation. Note that the first DA reported in 1928 by Diels and Alder was the reaction between Cp **1** and maleic anhydride **2**, a strong electrophile, ω = 3.24 eV [1]. The inclusion of an electron-releasing (ER) group at the 1-position of Cp **7** markedly increases the nucleophilicity *N* index of the corresponding Cp derivative, between 3.62 eV (**12**, R = Me) and 4.65 (**14**, R = NMe**_2_**) eV. Consequently, it is expected that the DA reactions between these Cp derivatives and the acrolein:LA complexes will have a very strong polar character.

When a nucleophile approaches an electrophile along a polar process, some electron density is transferred from the nucleophilic to the electrophilic species. As a consequence, while the nucleophile loses some electron density, the electrophile gains it. These changes demand a reorganization of the electron density in both frameworks. In this context, the nucleophilic $P_{k\;}^{-}$ and electrophilic $P_{k\;}^{+}$ Parr functions [32] have shown to be the most accurate and insightful tools for the study of the electron density redistribution along a polar process, being widely used for the analysis of the local reactivity in polar processes. The electrophilic $P_{k\;}^{+}$ Parr functions of acrolein **7** and complexes **7-BH_3_** and **7-AlCl_3_**, and the nucleophilic $P_{k\;}^{-}$ Parr functions of Cp **1** and **13** are given in Figure 3.

Analysis of the electrophilic $P_{k\;}^{+}$ Parr functions of acrolein **7** indicates that the C5 carbon is the most electrophilic center of this molecule,$\;P_{k\;}^{+}$ = 0.52. In fact, this position is twice activated than the carbonyl C7 carbon,$\;\;P_{k\;}^{+}$ = 0.27 (for atom numbering see Scheme 4). Coordination of the LA to the carbonyl O8 oxygen increases the electrophilic $P_{k\;}^{+}$ Parr functions at the carbonyl C7 carbon, but the C5 carbon remains as the most electrophilic center of these LA complexes. Note that these changes do not modify the regioselectivity as in these LA complexes as the C6 carbon in electrophilic deactivated.

Analysis of the nucleophilic $P_{k\;}^{-}$ Parr functions of Cp **1** indicates that the C1 and C4 are symmetrically activated. These carbons are the most nucleophilic center of Cp **1**,$\;\;P_{k\;}^{-}$ = 0.47. Interestingly, the inclusion of an ER–OMe at the C1 carbon does not alter the nucleophilic $P_{k\;}^{-}$ Parr function at the C4 carbon,$\;\;P_{k\;}^{-}$ = 0.47, but relocates the nucleophilic $P_{k\;}^{-}$ Parr function at the C1 and C2 carbons. Consequently, the C4 carbon becomes the most electrophilic center of **13**.

Consequently, along a P-DA reaction involving non-symmetric reagents, the most favorable regioisomeric channels will be those associated with the two-center interaction between the most nucleophilic center of the diene framework, the C4 carbon, and the most electrophilic center of the electrophilic ethylene, the C5 carbon, in complete agreement with the experimental outcomes, and the energetic analysis of the different competitive reaction paths (see Section 2.3) [33].

### 2.2. Analysis of the Effects of the LAs and the Substitution on Cp **1** in the Electronic Structure of the Reagents

In order to get information about how the LA catalysts and the substituents present in Cp **1** can modify the electronic structure of the diene and ethylene frameworks, an electron localization function [34] (ELF) and natural population analysis [35,36] (NPA) analysis of **7** and the LA complexes **7-BH_3_** and **7-AlCl_3_**, and of Cps **1** and **13** was performed. The populations of the most relevant ELF valence basins of these reagents are gathered in Table 2, while the ELF localization domains are represented in Figure 4. The ELF-based Lewis-like structures and natural atomic charges are shown in [Fig molecules-25-02535-ch002].

ELF of acrolein **7** shows the presence of two disynaptic basins, V(C5,C6) and V’(C5,C6), integrating a total of 3.48 e, associated with the C5−C6 double bond, one V(C6,C7) disynaptic basin, integrating 2.22 e, associated with a C6−C7 single bond, one V(C7,O8) disynaptic basin, integrating 2.36 e, associated with a C7−O8 single bond, and two monosynaptic basins, V(O8) and V’(O8), integrating a total of 5.19 e, associated with the two lone pairs of the O8 oxygen. Two interesting conclusions can be obtained from this ELF analysis: i) the ethylene C5−C6 framework presents an identical electronic topology than that of ethylene **5 [11]**, indicating that at the GS the carbonyl group does not polarize the ethylene framework; and ii) the very low population of the carbonyl C7−O8 bonding regions, which corresponds with a C−O single bond, agrees with the high population of the two O8 lone pairs, indicating the high polarisation of the carbonyl group towards the O8 oxygen.

Coordination of the B or Al to the carbonyl O8 oxygen of acrolein **7** does not produce any remarkable change in the electronic structure of the acrolein framework of the two LA complexes (see Figure 4). Only the C5–C6 bonding region is polarized towards the C6–C7, the C7–O8 bonding region being practically unaltered. Note that the changes in population of the disynaptic basins are lesser than 0.1 e. The most noticeable change is the redistribution of the electron density between the two O8 lone pairs. The absence of a V(O8,Metal) disynaptic basin at these LA complexes indicates that the metal centers of these LAs are not covalently bonded to the carbonyl O8 oxygen [37].

ELF topology of Cp **1** shows the presence of two pairs of disynaptic basins, V(C1,C2) and V′(C1,C2), and V(C3,C4) and V′(C3,C4), integrating around 3.47 e each pair, associated with the C1−C2 and C3−C4 double bonds of Cp **1**, and one V(C2,C3) disynaptic basin, integrating 2.17 e, associated with the C2−C3 single bond. It is interesting to note that the C1−C2 bonding region of Cp **1** is identical than that of the C5−C6 bonding region of acrolein **7**, indicating that the carbonyl group does not polarize the C5−C6 double bond of acrolein **7**. With the inclusion of the strong ER –OMe group on the C1 carbon of Cp **7** non-relevant changes in the electronic structure of **13** are observed. Only the population of the C1−C2 and C3−C4 bonding regions are increased below 0.1 e.

From these ELF topological analysis at the GS of the reagents we can conclude that neither the nature of the LA, nor the substitution of the Cp core substantially modify the electronic structure of the diene and ethylene frameworks.

Finally, an NPA of the natural charges at these selected compounds was made. The ELF-based Lewis-like structures together with natural atomic charges are given in [Fig molecules-25-02535-ch002]. As expected, at acrolein **7**, the carbonyl C7 carbon presents a high positive charge, +0.37 e, and the O8 oxygen presents a high negative charge, −0.52 e, as a consequence of the strong polarisation of the carbonyl C7−O8 bonding region. On the other hand, the C5 carbon presents a high negative charge, −0.35 e, as a consequence of the more electronegative character of the carbon nuclei than the two attached hydrogens. Coordination of the LA to the carbonyl O8 oxygen polarizes the C7−O8 framework, increasing the positive charge at the carbonyl C7 carbon. As a consequence, the ethylenic C5−C6 bonding region is some polarized towards the carbonyl group, decreasing slightly the negative charge at the C5 carbon. In spite of this, the C5 carbon remains negatively charged.

On the other hand, the NPA of the natural charge at Cp **7** indicates that the C2 and C3 carbons are more negatively charged than the C1 and C4 ones. The inclusion of the ER –OMe group at the C1 carbon breaks the symmetric electron density distribution found at Cp **1**. Now the C1 and C2 carbon are more negatively charge than the C3 and C4 ones.

From the analysis of the global and local CDFT reactivity indices made in Section 2.1 and the present analysis of the electronic structure at the GS of the reagents some appealing conclusions can be obtained: i) ELF analysis of the electronic structure of the reagents indicates that neither substitution on the diene or ethylene, nor the use of the LAs, modify the electronic structure of diene or ethylene framework participating in the DA reactions. In fact, the topology of the C5=C6−C7 framework of acrolein **7** is similar to that of the C1=C2−C3 framework of Cp **1**; ii) NPA analysis of acrolein **7** indicates that only the carbonyl C7 carbon is positively charged, while the ethylene C5 carbon is negatively charged. Coordination of the LA to the carbonyl O8 oxygen does not modify substantially these behaviours; iii) NPA analysis of Cp **1** indicates that the C2 and C3 carbon are more negatively charged than the C1 and C4 ones; iv) inclusion of an ER –OMe group at the C1 carbon of Cp breaks the symmetric electron density distribution of Cp **1**, making the C1 carbon of Cp **13** the most negatively charged carbon; v) consequently, analysis of the electronic structure at the GS of the reagents does not permit to explain the reactivity of these species participating in P-DA reaction; vi) analysis of the CDFT reactivity indices accounts for the participation of acrolein **7** and Cp **1** as electrophile and nucleophile in P-DA reactions, as well as the effects of the LA coordination to the electrophilic acrolein **7**, and the ER substitution on Cp **1**; vii) analysis of the electrophilic and nucleophilic Parr functions at the non-symmetric reagents accounts for the regioselectivity in these P-DA reactions. While the non-substituted C5 carbon of acrolein **7** is the most electrophilic center of this species, the non-substituted C4 carbon of Cp **1** is the most nucleophilic center of this diene; and viii) the present analysis supports the earlier observation that the electrophilicity and nucleophilicity cannot be related only with charges at the GS of the molecules, but with the propensity of changes on electron density resulting of the GEDT along a polar reaction [38].

### 2.3. Study of the P-DA Reactions Between Cp **1** and the Series of LA Complexes

In order to shed light on the role of the LA catalysts on the DA reactions between Cp **1** and acrolein **7**, the P-DA reactions of Cp **1** and five selected LA complexes of increased electrophilic character were first studied (see Table 1). For a comparative analysis, the P-DA reactions between Cp **1** and acrolein **7** was also studied; the corresponding results are given in the Appendix A. Due to the non-symmetry of these LA complexes, two stereoisomeric reaction paths, the *endo* and the *exo*, are feasible (see Appendix A). Analysis of the stationary points involved in these P-DA reactions indicates that they take place along a one-step mechanism. Consequently, the reagents, one molecular complex (MC), **MC-LA**, two stereoisomeric TSs, **TSn-LA** and **TSx-LA**, and the corresponding cycloadducts (CA), **CAn-LA** and **CAx-LA**, were localized and characterized (Scheme 5). Relative gas phase electronic energies are given in Table 3. Total energies are given in Appendix A.

An exploration of the reaction paths between the separated reagents and the TSs allowed finding a series of MCs in which the two reagents are close in a parallel rearrangement. From these MCs, those associated to the *endo* reaction paths were selected as the energy reference. These **MC-LAs** are between 1.9 (**MC-AlMe_3_**) and 4.2 (**MC-AlCl_3_**) kcal·mol^−1^ more stable than separated reagents, being minimum in their corresponding potential energy surfaces (PESs). The activation energies associated to the two stereoisomeric reaction paths range from 13.0 to 5.2 kcal·mol^−1^ for the *endo* TSs, and from 13.0 to 5.6 kcal·mol^−1^ for the *exo* ones. The five P-DA reactions are exothermic between 13.8 and 14.8 kcal·mol^−1^.

Some appealing conclusions can be obtained from these energy results: i) the activation energies associated to these P-DA reactions are between 6.0 and 13.8 kcal·mol^−1^ lower than that associated with the P-DA reaction between Cp **1** and acrolein **7** (see Appendix A); ii) a good correlation between the electrophilicity ω index of the LA complex at the corresponding activation energy can be established. The more electrophilic the LA complex is, the more rapid the P-DA; iii) these P-DA reactions are low *endo* stereoselective; the *endo* TSs are found between 0.2 and 0.4 kcal·mol^−1^ lesser energetic than the *exo* ones; and iv) these P-DA reactions are exothermic between 13.8 and 14.7 kcal·mol^−1^. Consequently, coordination of the LA to the carbonyl oxygen of acrolein **7** only modifies the kinetics of the reactions.

The geometries of two stereoisomeric TSs involved in the P-DA reactions of Cp **1** with the series of LA complexes are given in Figure 3. From the distance between the C4−C5 and C1−C6 interacting carbons at the two stereoisomeric TSs some appealing conclusions can be obtained: i) in all TSs, the C4−C5 distances involving the most C5 electrophilic carbon of the LA complexes is shorter than the C1−C6 ones; ii) the C4−C5 distances vary is the narrow range of 1.97–2.00 Å, while the C1−C6 distances vary between 2.64–2.88 Å. Consequently, all TSs correspond to high asynchronous single bond formation processes; iii) an increase of the C1−C6 distance is observed with the increase of the electrophilic character of the LA complex; iv) the *endo* TSs are slightly more advance and more asynchronous than the *exo* TSs; v) the C1−C6 distances is slightly shorter at the *endo* TSs than at the *exo* ones. This behaviour is a consequence of the most favorable electrostatic interactions appearing in the zwitterionic *endo* TSs than at the *exo* ones; and vi) in spite of the high asynchronous character of **TSn-AlCl_3_**, intrinsic reaction coordinate (IRC) analysis of this TS indicates that it is associates to a one-step mechanism.

The polar nature of these LA catalyzed DA reactions was evaluated by computing the GEDT at the corresponding TSs. Reactions with GEDT values of 0.0 e correspond to non-polar processes, while values higher than 0.2 e correspond to polar processes. The computed GEDT values at the corresponding TSs varies from 0.27 e at **TSn-BH_3_** to 0.39 e at **TSn-AlCl_3_**, indicating the high polar character of these DA reactions (see Figure 5). Note that the GEDT at the *endo*
**TSn** associated to the uncatalyzed DA reaction is 0.15 e (see Appendix A). A slightly higher GEDT is found at the *endo* TSs that at the *exo* ones, as consequent of a more favorable electrostatic interaction at the zwitterionic *endo* TSs.

As expected, a very good linear correlation between the activation energies and the GEDT computed at the six TSs is found, R^2^ = 0.94 (see Figure 6). The values associated to the 32CA reaction of Cp **1** with ethylene **5** and acrolein **7** are also included. When the non-polar DA reaction with ethylene **7** and the most P-DA reaction with **7-AlCl_3_** are excluded, the R^2^ reaches the value of 0.99. The slope of the linear correlation, −46.8 (not shown in Figure 6), can be associated with the extent of activation energy decrease in these P-DA reactions with the increase of the GEDT taking place at the TS.

### 2.4. Study of the P-DA Reactions Between Substituted Cps **12**–**14** and Complex **7-BF_3_**

After to study the effect of the increase electrophilicity of acrolein **7** with the coordination to a LA in the before section, the role of the nucleophilic activation of the diene on the activation energy and selectivities was explored. To this end, the P-DA reactions of three 1-subtituted Cps, **12** (R = Me), **13** (R = OMe), and **14** (R = NMe_2_), of increased nucleophilic character (see Table 1), with complex **7-BF_3_** were studied. Because of the non-symmetry of these 1-subtituted Cps, together with the *endo*/*exo* stereoisomeric reaction paths, two regioisomeric reaction paths are feasible (see Appendix A). Consequently, the four competitive reaction paths were explored (see Scheme 6 and Scheme 7). Interestingly, due to the presence of the strong ER –OMe and –NMe_2_ groups in Cps **13** and **14**, respectively, allowing the stabilization of the corresponding zwitterionic intermediates, a two-step mechanism for the most favorable *ortho* regioisomeric reaction paths were found. Consequently, two TSs and one zwitterionic intermediate were located and characterized for the *ortho* regioisomeric reaction paths of **13** and **14**. Relative gas phase electronic energies are given in Table 4. Total energies are given in Appendix A.

Some appealing conclusions can be obtained from the relative energy given in Table 4: i) the most favorable reactive channels corresponds to *endo* channels via **TSon-C-BF_3_** or **TS1on-X-BF_3_**; ii) a strong reduction of the activation energy is found with the increase of the nucleophilic character of the Cp derivative. In fact, **TS1on-O-BF_3_** and **TS1on-N-BF_3_** are found below reagents. This behaviour demands the characterisation of the corresponding MCs, **MC-O-BF_3_** and **MC-N-BF_3_**, in order to obtain positive activation energies; ii) while the P-DA reactions of Cps **12** and **14** presents low *endo* selectivity as **TS1ox-X-BF_3_** are 0.9 and 0.7 kcal·mol^−1^ higher in energy than **TS1on-X-BF_3_**, respectively, the P-DA reactions of the methoxy derivative **13** presents a high *endo* selectivity as **TS1ox-O-BF_3_** is 2.7 kcal·mol^−1^ higher in energy than **TS1on-O-BF_3_**; iii) these P-DA reactions presents complete regioselectivity as **TSmn-X-BF_3_** are 5.7 (**12**), 14.1 (**13**) and 23.4 (**14**) kcal·mol^−1^ higher in energy than **TS1on-X-BF_3_**. The regioselectivity of these P-DA reactions increases notably with the increase of the polar character of the reaction (see later). Interestingly, while the activation energies associated to **TS1on-O-BF_3_** and **TS1ox-N-BF_3_** decrease with the increase of the nucleophilic character of the Cp derivative, the activation energies associated to **TSmn-O-BF_3_** and **TSmx-N-BF_3_** increase; iv) formation of the zwitterionic intermediates is exothermic by 6.6 (**ZWon-O-BF_3_**) and 20.9 (**ZWon-N-BF_3_**) kcal·mol^−1^; v) in spite of the high stabilization of the zwitterionic intermediates **ZWon-O-BF_3_** and **ZWox-O-BF_3_**, the low activation energies associated with the ring closure of these species, lesser that 1.3 kcal·mol^−1^, makes they unobservable. Consequently, in gas phase there are not any difference between the *two-stage one-step* mechanism [39] and the two-step one associated with the P-DA reactions of Cp **1** and Cp **13** with complex **7-BF_3_** (see Section 2.7 and Section 2.8).

The geometries of stereoisomeric TSs and intermediates involved in the P-DA reactions of Cp **13** with complex **7-BF_3_** are given in Figure 7, while those associated with the P-DA reactions of Cps **12** and **14** are given in Appendix A. The lengths of the C−C forming bonds are given in Table 5. Only the parameter associated with the P-DA reaction of Cp **13** will be herein commented on. From the distances between the C4−C5 and C1−C6 interacting carbons at the TSs and intermediates shown in Figure 5 some appealing conclusions can be obtained: i) the most favorable **TS1on-O-BF_3_** and **TS1ox-O-BF_3_** are slightly more delayed and more asynchronous than **TSn-BF_3_** and **TSx-BF_3_**; ii) at **ZWon-O-BF_3_** and **ZWox-O-BF_3_** the C4−C5 distances, 1.632 and 1.619 Å, respectively, indicate that the C4−C5 single bond has been already formed [13], while the C1−C6 distances remain larger than 2.56 Å. Note that while the length of C−C single bond at numerous organic compounds is 1.54 Å, that at reaction intermediates is slightly longer, 1.6 Å; iii) at the TSs associated with the ring closure, the C1−C6 distances are 2.185 Å at **TS2on-O-BF_3_** and 2.236 Å at **TS2ox-O-BF_3_**. These distances indicate that the formation of the news C−C single bond does not begun yet [13]; iv) the C1−C5 and C4−C6 distances at the regioisomeric TSs, 1.935 and 2.514 Å at **TSmn-O-BF_3_** and 1.934 and 2.582 Å **TSmx-O-BF_3_**, respectively, indicate that these TSs are more advanced and less asynchronous than the regioisomeric **TS1on-O-BF_3_** and **TS1ox-O-BF_3_**.

The GEDT values at the TSs associated with the nucleophilic attacks of the Cps **12–14** on the complex **7-BF_3_** ranges for 0.35 e (**TSmn-C-BF_3_**) to 0.41e (**TS1on-O-BF_3_**), while at the zwitterionic intermediates the corresponding values are found between 0.55 e (**ZWox-O-BF_3_**) to 0.79 e (**ZWon-N-BF_3_**). Some appealing conclusions can be obtained from the GEDT values given in Table 5: i) The high GEDT values found in these LA catalyzed reactions, higher than 0.35 e, account for the high polar character and low activation energies, lesser than 5 kcal·mol^−1^, associated with these P-DA reactions; ii) The GEDT found at the most favorable **TS1on-N-BF_3_**, 0.40 e, is slightly lower than that at **TS1on-O-BF_3_**, 0.41 e. This finding is a consequence of the fact that the GEDT values depend on the nucleophilic and electrophilic behaviours of the two reagents, but also of the distance between the two interacting frameworks. Note that the C4−C5 distances at the two TSs are 2.111 at Å **TS1on-O-BF_3_** and 2.328 at Å **TS1on-N-BF_3_**; i.e., the more advanced TS, the higher the GEDT is. Note that the maximum of GEDT along the reaction path is found at the ZW intermediates: 0.60 e at **ZWon-O-BF_3_** and 0.79 e at **ZWon-N-BF_3_**. At these intermediates, which present a similar C4−C5 distances, ca 1.60 Å, the GEDT values account for the more nucleophilic character of Cp **14** than Cp **13** (see Table 1); and iii) The GEDT values at the most favorable *endo* TSs and intermediates are slightly higher than that at the *exo* ones. The favorable electrostatic interactions appearing in the *endo* approach mode favors the GEDT and diminishes the energy of the *endo* TS. Interestingly, at the more polar *endo* structure, the C1−C6 distance is larger; iv) at the very unfavorable regioisomeric *meta* TSs, the GEDT is only slightly lower than that at the *ortho* TSs, but energetically are more unfavorable, even to the *endo* TS associated with P-DA between Cp **1** and complex **7-BF_3_** (see Table 5). This finding that was already highlighted in 2010, supports the concept of GEDT given in 2014 [13], since the electron density fluxes from the nucleophile to the electrophile, being not much dependent of the orientation of the approach of the reagents. As earlier was proposed, this finding goes against the arrows model for the “electron transfer” given in all organic textbooks.

An appealing question arises from the present GEDT analysis: why the activation energy decreases notably with the increase of GEDT at the more favorable *ortho* regioisomeric reaction paths, but it increases with the increase of GEDT at the more unfavorable *meta* regioisomeric reaction paths?

To answer this question, we need to consider two pertinent findings. First, based on the ELF topological analysis of the bonding changes along the reaction path of numerous organic reactions, in 2014 Domingo proposed a model for the C−C bond formation in organic reactions, in which they are formed by the C-to-C coupling of two *pseudoradical* centers created along the reaction paths [13]. This model was applicable even in ionic Diels–Alder reactions. In 2013, Domingo had already proposed the Parr functions [33], as the changes in spin electron density after the transfer of one electron from the nucleophile to the electrophile, to explain the local reactivity in polar reactions. Second, the high activation energy found in non-polar reactions was associated with the energy demanded for the depopulation of the C−C double bonds, which is required for the formation of the two aforementioned *pseudoradical* centers [11]. Thus, the acceleration found in polar reactions was associated to the favorable electronic effects caused by the electron density transfer in the bonding changes [38].

Figure 8 shows the ELF localization domains of the regioisomeric **TS1on-O-BF_3_** and **TSmn-O-BF_3_** involved in the P-DA reaction between Cp **12** and complex **7-BF_3_**. ELF of **TS1on-O-BF_3_** shows the presence of two monosynaptic basins, V(C4) and V(C5), integrating 0.43 and 0.15 e, respectively, associated to the C4 and C5 *pseudoradical* centers demanded for the subsequent C4−C5 single bond formation. On the other hand, ELF of **TSmn-O-BF_3_** also shows the presence of two monosynaptic basins, V(C1) and V(C5), integrating 0.63 and 0.44 e, respectively, associated to the C1 and C5 *pseudoradical* centers demanded for the subsequent C1−C5 single bond formation.

Thus, while the C4 and C5 *pseudoradical* centers present at the more favorable **TS1on-O-BF_3_** correspond with the most nucleophilic and the most electrophilic centers of Cp **13** and complex **7-BF_3_**, respectively, being their formation favored by the GEDT (see Parr functions in Figure 1), the formation of the C1 *pseudoradical* center at **TSmn-O-BF_3_** is not favored by the GEDT as it mainly directs the electron density towards the C4 carbon.

Consequently, along the most favorable regioisomeric reaction path, the GEDT facilitates the bonding changes demanded to reach **TS1on-O-BF_3_**, such as the Parr functions predict, but at the regioisomeric **TSmn-O-BF_3_**, where the two reacting molecules are approximated in a contrary orientation, the formation of the C1 *pseudoradical* center is not facilitated by the GEDT, as it mainly directs the electron density to the C4 carbon. Therefore, **TSmn-O-BF_3_** is energetically more unfavorable [33]. When more polar the DA reaction, higher the difference energy between two the regioisomeric paths is. This behaviour allows explaining the increase of the regioselectivity in LA catalyzed DA reactions.

### 2.5. Study of the Solvent Effects in the P-DA Reaction Between Cp **13** and Complex ***7-BF_3_***

Due to the high polar character of the species involved in the P-DA reactions of Cp and the series of LA complexes, solvent effects of dichloromethane (DCM) in the P-DA reaction between Cp **13** and complex **7-BF_3_**, which involves the formation of zwitterionic intermediates, were studied. Relative electronic energies in DCM are given in Table 6, while the geometries of the TSs are given in Figure 7. Total energies in DCM are given in Appendix A.

Inclusion of solvents effects stabilize the gas phase stationary points between 9 and 18 kcal·mol^−1^. The more stabilized species are the TSs and intermediates due to their high polar character. In DCM **TS1on-O-BF_3_** is found 2.3 kcal·mol^−1^ below the separated reagents, but if the formation of MC is considered, the activation energy with respect to this species becomes 1.4 kcal·mol^−1^. Solvent effects decrease the *endo* selectivity because as higher solvation of **TS1ox-O-BF_3_**, dipole moment 11.1 debye, than **TS1on-O-BF_3_**, dipole moment 10.4 debye. Now, *exo*
**TS1ox-O-BF_3_** is found 1.9 kcal·mol^−1^ above *endo*
**TS1on-O-BF_3_**. As a consequence of the high zwitterionic character of **ZWon-O-BF_3_** and **ZWox-O-BF_3_**, in DCM these species are found 12.7 and 10.9 kcal·mol^−1^ below the separated reagents. This strong stabilization increases the activation energies associated with the ring closure via **TS2on-O-BF_3_** and **TS2ox-O-BF_3_** to 7.4 and 5.7 kcal·mol^−1^, respectively. Solvent effects decrease the exothermic character of the reactions in ca. 2 kcal·mol^−1^ because a large solvation of the reagents than cycloadducts.

Interestingly, solvent effects of DCM reduce slightly the regioselectivity because a larger solvation of the regioisomeric **TSmn-O-BF_3_** and **TSmx-O-BF_3_** than **TS1on-O-BF_3_** and **TS1ox-O-BF_3_**. In spite of that, the reaction is completely *ortho* regioselective as the most unfavorable *meta/endo* regioisomeric **TSmn-O-BF_3_** is located 12.8 kcal·mol^−1^ above the *ortho/endo*
**TS1on-O-BF_3_**.

### 2.6. Thermodynamic Analysis of the P-DA Reactions Between Cp **13** and Complex **7-BF_3_**

Finally, the thermodynamic data of the P-DA reaction of Cp **13** and complex **7-BF_3_** as a representative LA catalyzed DA reaction were analyzed. The relative enthalpies and Gibbs free energies are given in Table 6. Total thermodynamic data are given in Appendix A. A representation of the enthalpy and Gibbs free energy profiles associated with the four competitive reaction paths are given in Figure 9.

Inclusion of the thermal correction to the electronic energies decreases the relative enthalpies between 1.1 and 3.1 kcal·mol^−1^. A low incidence has in the relative enthalpies of the TSs associated to the nucleophilic attack of Cp **13** and complex **7-BF_3_** which decrease between 1.1 and 1.2 kcal·mol^−1^ with respect to the electronic energies in DCM.

The inclusion of the thermal correction and entropies to enthalpies increase the relative Gibbs free energies between 11.3 and 14.2 kcal·mol^−1^ as consequence of the unfavorable activation entropy associated to these bimolecular processes, between 37.5 and 52.2 cal·mol^−1^·K^−1^. The activation entropies associated with the nucleophilic attack of Cp **13** to complex **7-BF_3_** are found in a short range, −45.0 and -45.8 cal·mol^−1^·K^−1^.

The activation Gibbs free energy associated to the P-DA reaction of Cp **13** with complex **7-BF_3_** rise to 11.3 kcal·mol^−1^, while formation of **CAon-O-BF_3_** become endergonic by 8.4 kcal·mol^−1^.

### 2.7. BET Study of the P-DA Reaction Between Cp **1** and Complex **7-BF_3_**

A bonding evolution theory [40] (BET) study the *endo* reaction path associated with the P-DA reaction between Cp **1** and complex **7-BF_3_**, as reaction model of this series of LA catalyzed DA reactions, was performed in order to understand the bonding changes along the reaction path, and thus to establish the molecular mechanism of these reactions. The detailed BET study of this P-DA reaction is given in Appendix A. The attractor positions of the ELF basins for the structures involved in the bond formation processes are represented in Figure 10.

From the BET analysis of the P-DA reaction between Cp **1** and complex **7-BF_3_** some noteworthy conclusions can be drawn: i) this P-DA reaction takes place along ten different phases; ii) formation of the first C4−C5 single bond takes place at *Phase VII* at a C−C distance of 1.95 Å, by sharing the non-bonding electron densities of the pair of C4 and C5 *pseudoradical* carbons [41] (see structures **S6** and **S7** in Figure 7); iii) formation of the second C1−C6 single bond takes place at the beginning of the last *Phase X*, at a C−C distance of 2.09 Å, by sharing the non-bonding electron densities of the two C1 and C6 *pseudoradical* carbons (see structures **S9** and **S10** in Figure 7); iv) formation of the second C1−C6 single bond begins when the population of the first C4−C5 single bond has reached 93% of its electron density in **CAn-BF_3_**. This means that this reaction path is associated to a *two-stage one-step* mechanism [39]; v) the maximum of GEDT proceeds along *Phase VII*, ca. 0.36 e. This very high GEDT is a consequence of the strong nucleophilic character of Cp **1** and the strong electrophilic character of complex **7-BF_3_**; vi) similar to the non-polar DA reactions, the activation energy of this LA catalyzed DA reaction can mainly be associated to the continuous depopulation of the C1−C2, C4−C4, and C5−C6 bonding regions along *Phases I-IV*, which is demanded for the subsequent creation of the two C4 and C5 *pseudoradical* carbons. Thus, the presence of the LA favors the depopulation of the C−C double bonds present at the reagents.

### 2.8. Stablishing the Similarity Between Two-Stage One-Step and Two-Step Mechanisms in LA Catalyzed DA Reactions

In 2008 Domingo proposed the concept of *two-stage one-step* mechanisms to describe those reactions in which formation of the two C−C single bonds takes place in a non-concerted process but in a single kinetic step [39]. At the high asynchronous TSs associated to these one-step reactions only one C−C single bond is being formed along a two-center interaction, while the formation of the second C−C single bond begins at the end of the reaction path, when the formation of the first C−C single bond is practically formed [39].

In order to stablish the similarity between the *two-stage one-step* and the two-step mechanisms, a comparative analysis of the ELF of the TSs and intermediate involved in the *endo* reaction path of the two-step P-DA reaction between Cp **13** and complex **7-BF_3_**, and some selected structures of the IRC of *endo* reaction path of the *two-stage one-step* P-DA reaction between Cp **1** and complex **7-BF_3_**, was performed. Figure 11 shows the position of the ELF attractors of **TS1on-O-BF_3,_ ZWon-O-BF_3_** and **TS2on-O-BF_3_**. ELF analysis of the *endo* reaction path of the *two-stage one-step* P-DA reaction between Cp **1** and complex **7-BF_3_** is given in Appendix A.

ELF of **TS1on-O-BF_3_** shows the presence of two monosynaptic basins, V(C4) and V(C5), integrating 0.43 and 0.17 e, respectively (see Figure 11). As expected, these monosynaptic basins appear at the most nucleophilic center of Cp **13**, the C4 carbon, and at the most electrophilic center of complex **7-BF_3_**, the C5 carbon. ELF of **TS1on-O-BF_3_** is closely to that of **TSn-BF_3_** (see Appendix A). The population of V(C4) and V(C5) monosynaptic basins is slightly higher in **TSn-BF_3_**, 0.48 and 0.33 e, respectively, as this TS is more advanced than **TS1on-O-BF_3_**; the C4−C5 distance at these TSs is 2.111 Å at **TS1on-O-BF_3_** and 1.977 Å at **TSn-BF_3_**. Consequently, **TS1on-O-BF_3_** and **TSn-BF_3_** are associated to the same chemical process; i.e., the nucleophilic attack of Cp **1** or Cp **13** to complex **7-BF_3_**. Sometimes, these high asynchronous TSs have been associated to a Michael addition [42].

ELF of **ZWon-O-BF_3_** shows the disappearance of V(C4) and V(C5) monosynaptic basins, and the appearance of a new V(C4,C5) disynaptic basin integrating 1.59 e, indicating the formation of the first C4−C5 single bond. ELF of **ZWon-O-BF_3_** is closely to that of the last structure of *Phase VII*, in which the V(C4,C5) disynaptic basin integrates 1.58 e. The length of the C4−C5 single bond at these species is 1.632 Å at **ZWon-O-BF_3_** and 1.624 Å at the last structure of *Phase VII*.

Finally, ELF of **TS2on-O-BF_3_** shows the presence of two monosynaptic basins, V(C1) and V(C6), integrating 0.26 and 0.44 e, respectively. At this TS, the population of the new V(C4,C5) disynaptic basin has been increased by only 0.09 e. ELF of **TS2on-O-BF_3_** is also closely to that of the structure of **S9** (see Figure 11), in which the population of V(C1) and V(C6) monosynaptic basins is slightly lower, 0.18 and 0.33 e, respectively; the C1−C6 distance at these species is 2.185 Å at **TS1on-O-BF_3_** and 2.229 Å at **S9**.

Consequently, this ELF comparative analysis allows establishing that the electronic structures and geometries of **TSn-BF_3,_** and the two selected points of the IRC defining the *two-stage one-step* mechanism are very closely to those of **TS1on-O-BF_3,_ ZWon-O-BF_3_** and **TS2on-O-BF_3_**, asserting a similar bonding changes along the two reaction mechanisms.

Thus, where is the difference between both mechanisms? Both reactions are polar processes associated to the nucleophilic attack of the C4 carbon of Cp frameworks to the C5 carbon of complex **7-BF_3_**. The GEDT increases along this polar process, reaching the maximum value at **ZWon-O-BF_3_**, 0.60 e, and **S9**, 0.32 e. The high GEDT found at the zwitterionic intermediate is a consequence of the presence of the strong ER -OMe group at the Cp framework that not only favors the GEDT, but also stabilize thermodynamically the corresponding intermediate, thus becoming as a stationary point along the reaction path, a behavior that is not so effective at the structure **S9**. Note that relative energies of **ZWon-O-BF_3_** and **S9**, with respect to the separated reagents are −6.6 and 6.6 kcal·mol^−1^.

So, the thermodynamic stabilization of some structure after the formation of the first C−C single bond along the reaction path is able to change the molecular mechanism from a *two-stage one-step* mechanism to two-step one. In that study, the change of the weak ER methyl group in Cp **12** by a strong ER -OMe group in Cp **13** changes the molecular mechanism from one-step to a two-step one. Sometimes, the use of polar solvent in the geometry optimizations, the use of DFT functional such as the M0-6X, which favors the polar process, of the use of larger basis sets, able to stabilize a zwitterionic intermediate, can also changes the molecular mechanism from a *two-stage one-step* to a two-step one, but this behaviour has no significance as the chemical characteristics of these reactions, i.e., reactions rates and selectivities are generally established at the TSs associated with the initial nucleophilic/electrophilic interaction.

## 3. Conclusions

The role of the metal-based LA catalysts on the P-DA reactions of a series of substituted Cps of increased nucleophilicity towards acrolein **7** has been analyzed within MEDT using DFT calculations at the B3LYP/6-311G(d,p) computational level. The role of the LAs in the activation energies, and in the regio- and stereoselectivity of these P-DA reactions have been analyzed.

Analysis of the electrophilicity ω index of the corresponding LA complexes shows that there is a relationship between the increase of the electrophilic character of the LA complex and the acidic character of the LA salts. This behaviour allows the participation of the corresponding LA complexes in P-DA reactions with low activation energies. ELF and NPA analyses of the LA complexes show that coordination of the LA salt to the carbonyl oxygen of acrolein **7** does not substantially modify the electronic structure at the GS of this ethylene derivative.

The study of the P-DA reactions of Cp **1** with the series of the five LA complexes of increased electrophilic character shows that there is a clear correlation between the reduction of the activation energies of these P-DA reactions and the increase of the polar character of the reaction measured by analysis of the GEDT at the corresponding TS. These P-DA reactions take place with low *endo* stereoselectivity, but a complete regioselectivity resulting of the most favorable two-center interaction between the most electrophilic carbon of acrolein **7** and one of the two end carbons of the diene system of Cp **1**.

A similar trend is observed when the nucleophilic character of Cp **1** is increased with the inclusion in the diene system of a ER group. In the cases of the methoxy and the amino substituted Cps **13** and **14**, the presence of these strong -OMe and -NMe_2_ ER groups in the diene allow the stabilization the positive charges that is developing in the Cp framework, changing the mechanism from a one-step to a two-step one via formation of a zwitterionic intermediate.

The inclusion of solvent effects of DCM does not modify substantially the gas phase result. The inclusion of the thermal corrections to the electronic energies increases the activation enthalpies by only a 1.0 kcal·mol^−1^. On the other hand, the inclusion of the TΔS factor to the enthalpies increases the activation Gibbs free energies by ca. 12.0 kcal·mol^−1^ as a consequence of the bimolecular nature of the DA reaction.

BET analysis of the P-DA reaction of Cp **1** with the complex **7-BF_3_** indicates that this DA reaction takes place through a *two-stage one-step* mechanism characterized by el nucleophilic attack of the C1 carbon of Cp **1** on the most electrophilic center of the complex **7-BF_3_**, the β-conjugated C5 carbon. Formation of the second C−C single bond takes place at the end of the reaction path when the first C−C single bond is already practically formed. This behavior allows explaining the total regioselectivity of these P-DA reactions.

An ELF comparative study of the electron density of the TSs and the zwitterionic intermediate involved in the two-step P-DA reaction between Cp **13** and complex **7-BF_3_** with some selected points of the IRC of the *two-stage one-step* P-DA reaction between Cp **1** and complex **7-BF_3_** shows the great similitude in bonding changes in both LA catalyzed DA reactions. The unique difference between both mechanisms is found in a thermodynamic question: the presence of the strong ER -OMe group in Cp **13** is able to stabilize the corresponding zwitterionic intermediate, thus changing the molecular mechanism. However, as the second step of these two-step mechanism has unappreciable barrier, this change of mechanism has not any chemical relevance.

This MEDT study makes it possible to establish the role of the LA catalyst in P-DA reactions: coordination of the LA metal to acrolein **7** increases notably the polar character of these DA reactions. LA catalysts increase notably the regioselectivity of these P-DA reactions to favor only one of the two possible regioisomeric reaction paths. Thus, the favorable nucleophilic/electrophilic interaction taking place at the corresponding polar TSs, and not the decrease of the Pauli repulsions, such has Bickelhaupt recently proposed [24], are responsible for the high acceleration and complete regioselectivity of LA catalyzed DA reactions.

Organic chemists demand compressible theoretical models easy to handle, able to predict experimental organic reactions. Well established concept such as nucleophilicity and electrophilic are used in Organic Chemistry to explain chemical reactivity. Today, the easy computation of the electrophilicity ω and nucleophilicity *N* indices within the CDFT has become these theoretical indices as a powerful tool for the experimental organic chemists to understand, even predict chemical reactions [18].

## 4. Computational Methods

All stationary points were optimized using the B3LYP functional [43,44], together with the 6-311G(d,p) basis set [45]. The optimizations were carried out using the Berny analytical gradient optimization method [46,47]. The stationary points were characterized by frequency computations in order to verify that TSs have one and only one imaginary frequency. The intrinsic reaction coordinate (IRC) paths [48] were traced in gas phase in order to check and obtain the energy profiles connecting each TS to the two associated minima of the proposed mechanism, i.e., reactants and products, using the second order González-Schlegel integration method [49,50]. Solvent effects of DCM were taken into account by full optimization of the gas phase stationary points using the polarizable continuum model (PCM) [51,52] in the framework of the self-consistent reaction field (SCRF) [53,54,55]. Values of B3LYP/6-311G(d,p) enthalpies, entropies and Gibbs free energies in DCM were calculated with standard statistical thermodynamics at 273.15 K and 293.15 K and 1 atm [45].

The electronic structures of the stationary points were characterized by NPA [35,36], and by the topological analysis of the electron localization function (ELF) [34]. CDFT reactivity indices [17,18] were computed at the B3LYP/6-31G(d) level using the equations given in reference 18. The GEDT [13] was computed by the sum of the atomic charges (q) of the atoms belonging to each framework at the TSs; GEDT = Σq_f_.

All computations were carried out with the Gaussian 16 suite of programs [56]. ELF studies were performed with the TopMod program (Tamu, TX, USA) [57], using the corresponding B3LYP/6-31G(d) monodeterminantal wavefunctions and considering the standard cubical grid of step size of 0.1 Bohr. The molecular geometries and ELF basin attractor positions were visualized using the GaussView program (Wallingford, CT, USA) [58], while the ELF localization domains were represented by means of the Paraview software (Clifton Park, NY, USA) at an isovalue of 0.75 a.u. [59,60].

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
