# Peer review of "Unveiling the Lewis Acid Catalyzed Diels–Alder Reactions Through the Molecular Electron Density Theory"

_molecules, 2020, doi:10.3390/molecules25112535_

Round 1
Reviewer 1 Report
the authors offer a hudge MEDT study of the Lewis acids -catalyzed P-DA reactions by studying the contribution of different parameters in the mechanism and the regioselectivity of the reaction. For this, the authors highlight the contribution of the electrophilic / nucleophilic character, the role of LA on electronic distribution, etc. by studying the derivatives of cp and acrolein in the presence of LA. The study is supplemented by a study of the energies between different partners in order to determine the contribution of LA on the reaction path (GS, TS, CA).
Moreover the authoirs considered also the favorable regioisomeric path regarding the LA role, the solvation effect of the reagents by the solvent in the stabilization of the zwitterioninc intermediates and transition states, the thermodynamic data and the mechanism(s) of the reaction.
the study addresses all the parameters of the reaction and the role of Lewis acid to explain the kinetics and the regioselectivity of the reaction.

Author Response
Referee 1 made the suggestions on the manuscript itself.
Line 46, “mechanism” has been changed by “reaction path”.
Line 47, OK.
Line 77, the quotes finishes in line 80.
Line 107 the Lewis structures 9 and 11 has been corrected.
Line 188, the values of the global electrophilicity and nucleophilicity indices do not permit to estimate the polar character of the reaction, but the reactions of a strong electrophile and a strong nucleophile, have usually a strong polar character.
Line 213, yes, it is the delta conjugated position with respect to the methoxy substituent.
Line 214-218, the line space has been corrected.
Finally, all suggested changes in references have been done.
Reviewer 2 Report
The authors discuss an in-depth analysis of the effect of different Lewis acid catalyst on the polar Diels-Alder reaction between cyclopentadiene derivatives and acrolein-LA complexes as the dienophile. The discussion is elegantly split into different sections covering analyses. I must admit I am not an organic chemist, but I accepted the review as a materials engineer using among other DA chemistries in reversible polymer networks. The topic of this manuscript sparked my interest and has drawn me out of my comfort zone. I must commend the authors for making it possible for me to understand this interesting study. As I have no practical experience with DFT studies, I have limited my comments to the analyses of the results.
Abbreviations of LA, ER and TSs were not defined in the abstract. Also, at the first mention in the body of the text these abbreviations should be defined.
The novelty or added value of the current research should be highlighted.
I am not an organic chemist, but I am used to the terminology “dienophile”. I was surprised to only find it twice in this manuscript. Is this not a better terminology than “ethylene series”?
The numbering of the dienes and dienophiles needs to be performed as clear and as early as possible in the manuscript. Figure 1 presents the activation barriers for the reaction between Cp and many dienophiles, with their structures shown (very good). However, the paragraph discussion Figure 1 uses numbers for the dienophiles, that have not been assigned to structures. Conversely, Figure 2 shows numbers in the graph, while these numbers are only clarified in Scheme 4, several pages further. Please add the relevant numbers to the structures in Figure 1. It could even be worthwhile considering moving the paragraph starting at line 68 forward, such that Figure 1 and Figure 2 could fit on the same page or to fit the graphs next to each other in 1 figure.
Similarly, in line 82, the structure for tetrazine 8 never shown. Please, add an example of a reaction between a tetrazine and nitroethylene to Scheme 3 and refer to Scheme 3 in the text.
How is R² = 1 in Figure 2 if the linear regression doesn’t go through any of the data points?
For Figure 3 a clear numbering of the carbon atoms in the chemical structure(s) is required, as this is only available in Scheme S4, without proper reference to it. A drawing similar to the left part of Scheme S4 could be added to Figure 3 for clarification.
Section 2.1 gives an interesting overview and comparison of the electrophilicity and nucleophilicity of the dienes and dienophiles. It is mentioned that both the metal centre and the ions have an important influence on these properties. Is the set of complexes too narrow to already draw stronger conclusions here, e.g. comparing BH3 with BF3; AlMe3 with AlCl3 and TiCl4 with AlCl3?
On lines 329-330 the statement “coordination of the LA to the carbonyl oxygen of acrolein 7 only modifies the kinetics of the reactions” was made. Please, elaborate “only”.
In light of the previous comment, theoretically all Diels-Alder reactions are dynamically reversible, though the reverse retro DA reaction is not always observed or even practically attainable.
Small corrections:
- Line 50: too many spaces, between [6] and the comma, and after the comma
- The main sentence “that with acrolein 7 must be heated” of lines 69-69 does not make sense. Please adapt “the reaction between Cp 1 and acrolein 7 only proceeds at higher temperatures”
- Line 71: “or” i.o. “o”
- Line 111: no need to specify the journal and publication title of the reference. Year and author name suffice, along with the reference number. Combine references in line 113.
- Review all subscript numbers for chemical formulas, as the numbers are small number, but not subscripts, e.g. BH3 and ACl3o. BH3 and AlCl3
- Line 213: “becomes” i.o. “comes”
- Line 284: positively charged. “charged” is missing
- Line 286: “Cp 1” i.o. “Cp 7"
Author Response
Abbreviations of LA, ER and TSs were not defined in the abstract. Also, at the first mention in the body of the text these abbreviations should be defined.
R. The abbreviations have been defined.
The novelty or added value of the current research should be highlighted
R. The added value of the current MEDT study is highlighted at the end of the Introduction part: ”The main propose of this MEDT study is to shed light on all mechanistic details of these pertinent organic reactions which have acquired a great significance in the asymmetric synthesis with the use of chiral LA catalyst”.
I am not an organic chemist, but I am used to the terminology “dienophile”. I was surprised to only find it twice in this manuscript. Is this not a better terminology than “ethylene series”?
R. We thanks this observation. In our studies of P-DA reactions, in which we address about nucleophiles and electrophiles, we refer to the reagents as dienes and ethylene derivatives, which is the name of these organic compounds. Consequently, word “dienophile” has been removed in this revised version.
The numbering of the dienes and dienophiles needs to be performed as clear and as early as possible in the manuscript. Figure 1 presents the activation barriers for the reaction between Cp and many dienophiles, with their structures shown (very good). However, the paragraph discussion Figure 1 uses numbers for the dienophiles, that have not been assigned to structures. Conversely, Figure 2 shows numbers in the graph, while these numbers are only clarified in Scheme 4, several pages further. Please add the relevant numbers to the structures in Figure 1. It could even be worthwhile considering moving the paragraph starting at line 68 forward, such that Figure 1 and Figure 2 could fit on the same page or to fit the graphs next to each other in 1 figure.
R. We thanks this observation. The number of the relevant structures has been added to Figure 1. In addition, the formulae of the compound given in Figure 2 have been included in this figure.
Similarly, in line 82, the structure for tetrazine 8 never shown. Please, add an example of a reaction between a tetrazine and nitroethylene to Scheme 3 and refer to Scheme 3 in the text.
R. In agreement with the referee’s suggestion, a new Chart 1 with the structure of tetrazine 8 has been included in this revised version.
How is R² = 1 in Figure 2 if the linear regression doesn’t go through any of the data points?.
R. The R2=1.0 is given by the linear regression analysis given by the Excel program. Note that the four points are found practically in the line of the linear regression.
For Figure 3 a clear numbering of the carbon atoms in the chemical structure(s) is required, as this is only available in Scheme S4, without proper reference to it. A drawing similar to the left part of Scheme S4 could be added to Figure 3 for clarification.
R. In agreement with the referee’s suggestion, the atom numbering has been included in Scheme 4, and indicated in the text.
Section 2.1 gives an interesting overview and comparison of the electrophilicity and nucleophilicity of the dienes and dienophiles. It is mentioned that both the metal centre and the ions have an important influence on these properties. Is the set of complexes too narrow to already draw stronger conclusions here, e.g. comparing BH3 with BF3; AlMe3 with AlCl3 and TiCl4 with AlCl3?
R. We agree with the referee’s comment that the set of complexes too narrow, by it is well known that the electro-withdrawing character is F > H and Cl > Me, and that the electronegativity of Al is higher than that of Ti.
On lines 329-330 the statement “coordination of the LA to the carbonyl oxygen of acrolein 7 only modifies the kinetics of the reactions” was made. Please, elaborate “only”.
In light of the previous comment, theoretically all Diels-Alder reactions are dynamically reversible, though the reverse retro DA reaction is not always observed or even practically attainable.
R. In agreement with the referee`s comment we have included a new phrase in line 330 justifying that the LA catalyst does not affect to thermodynamic of the reaction. “Note that the exothermic character these LA catalyzed P-DA reactions are found in the narrow range of one kcal·mol-1 “
Small corrections:
- Line 50: too many spaces, between [6] and the comma, and after the comma
- R. The spaces have been removed.
- The main sentence “that with acrolein 7 must be heated” of lines 69-69 does not make sense. Please adapt “the reaction between Cp 1 and acrolein 7 only proceeds at higher temperatures”
- R. In agreement with the referee’s suggestion, the sentence has been changed.
- Line 71: “or” i.o. “o"
- R. The mistake has been changed
- Line 111: no need to specify the journal and publication title of the reference. Year and author name suffice, along with the reference number. Combine references in line 113.
- R. In agreement with the referee’s suggestion, the journal and the name of the publication have been removed in this revised version.
- Review all subscript numbers for chemical formulas, as the numbers are small number, but not subscripts, e.g. BH3 and ACl3o. BH3 and AlCl3
- R. In agreement with the referee’s suggestion, all subscript numbers for chemical formulas have been revised.
- Line 213: “becomes” i.o. “comes”
- R. The mistake has been changed.
- Line 284: positively charged. “charged” is missing
- R. The word “charged” has been included.
- Line 286: “Cp 1” i.o. “Cp 7"
- R. The mistake has been changed.